# PRO: Pseudo-label Regularized Optimization on Unlabeled Test Data

## Abstract

Web-scale foundation models like CLIP have impressive zero-shot capabilities on many downstream classification tasks, but they still underperform target domain-specific supervised classifiers. This inspired researchers to investigate adaptation strategies that take advantage of unlabeled data, often via pseudolabeling. However, previous methods for adaptation can be difficult to train; poor hyperparameter choices can result in catastrophic collapses in accuracy, and absent target labels, there is little to guide the search with. In this paper, we propose Pseudo-label Regularized Optimization (PRO), which addresses the collapses in test-time adaptation without any label peeking for hyperparameter tuning. On the 18 datasets addressed in our experiments PRO improves the accuracy of ViT-B-32 by 2.5% on average and in the best case by 6.1% from tuning the textual encoder. Our code is available at `https://github.com/anonWAEWA/PRO`.

## 1 Introduction

A common assumption in machine learning is that the test data and the train data are drawn i.i.d from the same distribution. Yet, due to natural changes in the real world, the i.i.d. assumption is often violated in practice (Schölkopf et al., 2021). The test data might originate from a distribution distinct from that of the training data, which often leads to a decline in model performance (Quionero-Candela et al., 2009; Torralba & Efros, 2011; Hendrycks & Dietterich, 2019a; Yao et al., 2022; Garg et al., 2023). Common practice is to adapt the model parameters to the test distribution by training on a suitable combination of unlabeled test data and labeled training data (Lee et al., 2013; Laine & Aila, 2017; Tarvainen & Valpola, 2017; Sohn et al., 2020). Unfortunately, in the age of generalist foundation models, this can be infeasible. As models grow in size, their training data can become prohibitively expensive to access. Additionally, it has become increasingly common to rely on models trained by other researchers (often by large companies), for which the training data is often unavailable, either because it is difficult to access (e.g. requiring crawling or downloading large datasets) or lacking details, e.g., GPT4 (OpenAI, 2023), CLIP (Radford et al., 2021). Regardless, these models are attractive to practitioners since they can be used for many tasks without collecting labeled data.

This introduces a dilemma. Deploying the model in a new setting using only unlabeled data is far easier than annotating a portion of this target data. At the same time, adapting the model in some way is likely to yield significant improvements. Therefore, practitioners may hope to use the unlabeled data from their specific problem to adapt their pretrained models. This setting is known as *test-time training* (TTT). A number of prior works have tried to identify good recipes to improve the performance of pre-trained deep networks under the test-time training setting (Sun et al., 2020; Liang et al., 2020; Liu et al., 2021). Common tricks include sharpening the model's predictions with a test-time surrogate loss and mean teacher schemes (Wang et al., 2021; Sinha et al., 2023). While test-time training is a generic paradigm, we focus our narrative around visual-language (VL) models, leaving consideration for more traditional models in App B. VL models like Contrastive Language-Image Pre-Training (CLIP) (Radford et al., 2021; Cherti et al., 2022) are often used by practitioners in a number of different tasks without fine-tuning, especially when access to any labeled data is lacking. The key limiting factor for the widespread adoption of test-time training has been a common failure mode thoroughly investigated by Zhao et al. (2023): naive adaptation of the model based on pseudo-labels increases its accuracy in the short term but eventually leads to catastrophic collapse. They point out that the performance of methods developed for test-time training often suffers from three

dependencies: (1) the hyperparameters of the method, (2) the type of pretrained models, and (3) the nature of the distribution shifts. The effects of these three issues on the performance of the test-time training approaches are challenging to quantify without access to supervision.

In this paper, we revisit the problem of the collapse that test-time training methods commonly suffer from (Zhao et al., 2023) and consider a number of heuristics proposed in prior works to address the collapse (Laine & Aila, 2017; Tarvainen & Valpola, 2017; Li et al., 2023). To identify promising approaches, we first adapt the linear classifier of the ViT-B-32 CLIP model to 14 different downstream datasets, while limiting ourselves to the setting where a single choice of hyperparameters needs to be applicable to all. We then propose Pseudo-label Regularized Optimization (PRO), an adaptation scheme that exploits a set of regularizations to test-time training to avoid collapses and enable significant improvement in the model's accuracy. Next, we assess the performance of PRO-CLIP on additional datasets, on tuning the textual tower of ViT-B-32, and on adapting the larger ViT-L-14-336 model. We find that, aside from decreasing the learning rate when tuning the textual tower, PRO-CLIP prevents collapses under all the ablations above and successfully improves the model performance without any posthoc tuning of the hyperparameters.

**Summary of contributions**:

- We demonstrate that naively applying semi-supervised learning (SSL) methods leads to severe decline in model accuracy in Sec. 3. We then identify the cause for these collapses, the convergence of model's predictions to few majority classes.

- In Sec. 5, we propose Pseudo-label Regularized Optimization (PRO), a scheme that combines ideas previously proposed in TTT and SSL settings to avoid collapse and enable learning, all without the need to tune the hyperparameter.

- We demonstrate in Sec. 6 that PRO does not suffer from the meta-overfitting problem that previous work pointed out (Zhao et al., 2023) by applying PRO to various settings while keeping the hyperparameters fixed.

## 2 RELATED WORKS

**Semi-supervised Learning** is a setting where both labeled source data and unlabeled target data are provided, and the goal is to leverage the additional unlabeled data during training to improve the model's performance. One common heuristic is *entropy minimization*, which follows the intuition that a good classifier should separate the unlabeled data well (Grandvalet & Bengio, 2004). A simple way to achieve this is to use the entropy of the model's prediction on unlabeled data as a surrogate loss for target distribution (Grandvalet & Bengio, 2004; Berthelot et al., 2019). The other common approach of training on pseudo-labels of the test data can be seen as another way to encourage entropy minimization (Lee et al., 2013; Sohn et al., 2020; Tarvainen & Valpola, 2017). When used in conjunction with a confidence threshold, these methods assign hard labels to high confidence predictions to train the model, thereby implicitly minimizing the entropy of the model's prediction. Other works based on pseduo-labels also explore the idea of *consistency regularization*, which encourages the output of the model to be invariant to small augmentations (Sohn et al., 2020; Laine & Aila, 2017). In a different direction, Berthelot et al. (2020) explore *information maximization*, which encourages the output of the model on the test data to be diverse in label marginals. All these ideas have been revisited at different points in the setting where only unlabelled data is available.

**Test-Time Training** focuses on adapting a pretrained model to a target distribution with access to only the unlabeled target data. The similarity between the settings of semi-supervised learning and test-time training allow many ideas to transfer from the former to the latter, e.g. *entropy minimization* (Wang et al., 2021), *consistency regularization* (Zhang et al., 2021), and *information maximization* (Liang et al., 2020; Li et al., 2023). However, under the restrictive settings of having no unlabelled data for the target domain, methods are susceptible to meta-overfitting to the particular architecture or distribution shift (Zhao et al., 2023). At the same time, the recent surge of interest in large foundation models such as CLIP (Radford et al., 2021), for which the training data are often inaccessible, increases the relevance of test-time training even further. CLIP is often used in practice because of its zero-shot capabilities, allowing users to not invest in labeling. Regardless, they may still want to improve the performance on their data, which is when test-time training becomes most appealing. The zero-shot capability of CLIP enables us to study the test-time training performance of a single

model on a very wide range of downstream tasks, each exhibiting its unique distribution shift. Despite this, test-time training for CLIP has remained curiously underexplored with the exception of the work by Li et al. (2023), which we found to be insufficient to address collapse. Instead, we will leverage additional approaches originally proposed in semi-supervised learning to show how we can avoid collapse and improve performance on multiple datasets.

## 3 PROBLEM SETTING

While test-time training is relevant for most machine learning models, we consider mainly the application of visual-language models (Radford et al., 2021; Cherti et al., 2022) on zero-shot classification tasks, a setting in which the shifts in distribution are possibly pronounced. This is because the same model can be deployed to multiple different tasks without fine-tuning, and each task may differ from the original training data in different ways (usually, it is more specific). In the main paper, we focus our exposition on the CLIP models and report experiments with a more traditional model trained to predict within a predetermined set of classes in Appendix. B.

Let $\mathcal{T}$ and $\mathcal{V}$ denote the textual and visual spaces. $f_{T,V} : \mathcal{T}, \mathcal{V} \to \mathbb{R}^d$ are the textual and visual encoders of a visual-language model trained on $\mathcal{D}_{\text{train}} = (\mathcal{T}_{\text{train}}, \mathcal{V}_{\text{train}})$. We use the VL model as a zero-shot image classification model $f_\theta : \mathcal{V} \to \Delta_K$, where $\Delta_K$ is a $K$-dimensional simplex, $K$ is the number of classes in a classification task, and $\theta$ are the tunable parameters. To employ the VL model for zero-shot image classification, we take the class names $c \in \mathcal{C}, |\mathcal{C}| = K$ of the classification task to generate prompts $t_c \in \mathcal{T}$ that describes the classes we wish to predict. For an image $v \in \mathcal{V}$, the VL model $f_\theta$ predicts probability of class $c$ as

$$p_c = \frac{e^{\cos(f_T(t_c), f_V(v))/T}}{\sum_{c' \in \mathcal{C}} e^{\cos(f_T(t_{c'}), f_V(v))/T}} \tag{1}$$

where $\cos$ denotes the cosine similarity and $T = 0.01$. As we are interested in the test-time training setting, the VL model is tasked to adapt to the new data $\mathcal{V}_{\text{test}}, \mathcal{V}_{\text{test}} \neq \mathcal{V}_{\text{train}}$ without having access to $\mathcal{D}_{\text{train}}$. Throughout this work, we freeze $f_V$ and consider tuning two different sets of parameters: (1) in one case, we use the pretrained textual tower $f_T(t_c)$ to initialize a linear classifier for subsequent tuning, and in the other case (2) we tune the entire textual encoder of the VL model.

A naive application of pseudo-labeling (Lee et al., 2013) in TTT setting often results in severe deterioration in the model's accuracy, a phenomenon that has also been reported by Zhao et al. (2023). Consider the pseudo-label based surrogate loss

$$\mathcal{L}^{(\text{p})}(\theta) = -\frac{1}{N} \sum_{i=1}^{N} \sum_{j=1}^{K} \mathbb{I}\left[j = \max_k p_\theta(y_k | x_i)\right] \log p_\theta(y_j | x_i) \tag{2}$$

and the standard surrogate loss in test-time training that encourages minimization of the entropy of model's prediction (Wang et al., 2021)

$$\mathcal{L}^{(\text{e})}(\theta) = \frac{1}{N} \sum_{i}^{N} \sum_{j}^{K} p_\theta(y_j | x_i) \log p_\theta(y_j | x_i).$$

If one updates a pretrained model to minimize the weighted sum of the two surrogate losses above

$$\mathcal{L}(\theta) = \mathcal{L}^{(\text{p})}(\theta) + \lambda_e \mathcal{L}^{(\text{e})}(\theta), \tag{3}$$

it is clear that one simple solution is to only predict one of the classes. Indeed, we observe that CLIP models converge to this trivial solution in practice as well.

**Collapse problem:** We illustrate the collapse in Figure 1 by adapting the linear classifier of

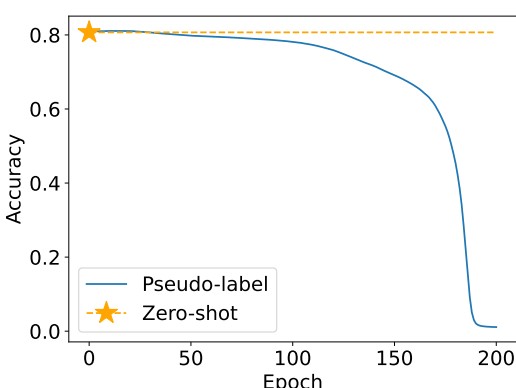

Figure 1: Collapse in model's accuracy when adapting the linear head of ViT-B-32 to the Food101 dataset based on pseudo-labels.

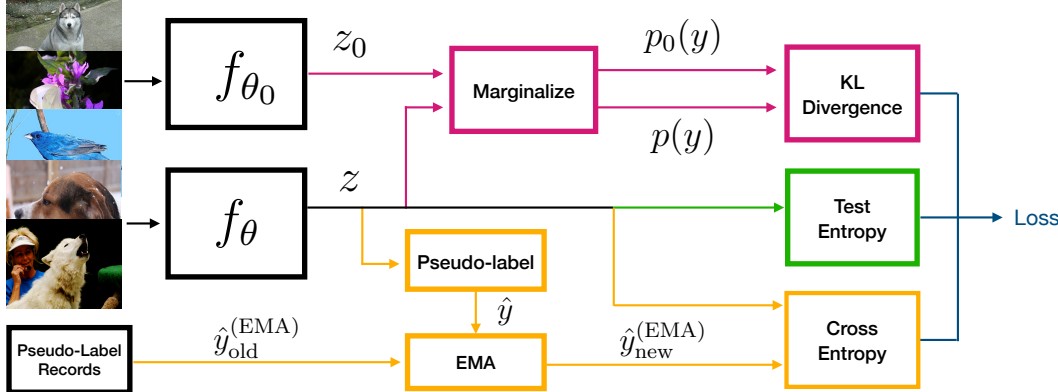

Figure 2: A sketch of PRO, where we guide test-time training with three main components: (1) consistency between current and initial average predictions, (2) entropy of model's prediction, and (3) pseduo-labels accumulated through exponential moving average (EMA) .

ViT-B-32 to the Food101 dataset (Bossard et al., 2014). After an initial increase, adapting on pseudo-labels results in performance much worse than that of the initial zero-shot model, eventually collapsing to the trivial solution of predicting a single class. Indeed, any spontaneous class imbalance in the pseudo-labels encourages the adapted models to predict the majority classes more often, which exacerbates the class imbalance and results in a vicious cycle. We observe similar collapses in 11 out of 14 datasets examined in this study from tuning the linear head of ViT-B-32. We remark that this problem was first recently observed by Zhao et al. (2023) on more traditionally trained models.

Clearly, if we had access to a labeled validation set, we could choose when to stop the optimization and tune other hyperparameters, but this is infeasible in the test-time training setup. After all, if we had a validation set, it is unclear why we wouldn't use it to fine-tune the model in the first place. Instead, we wish to change the optimization procedure so that it nicely converges without collapse and without needing to tune hyperparameters. In the next section, we explore methods to prevent these collapses and to learn from unlabeled data only.

## 4    APPROACHES TO ADDRESS COLLAPSES

Given the tendency for standard semi-supervised methods to collapse absent regularization provided by labeled source data, a natural solution is to introduce alternative regularizations to test-time training to avoid collapse. In this section, we enumerate a few techniques previously proposed in semi-supervsied and test-time training settings that may help stabilize test-time training (Laine & Aila, 2017; Tarvainen & Valpola, 2017; Wang et al., 2021; Li et al., 2023).

**Marginal Consistency (MC).** As demonstrated in the previous section, if the adaptation is solely based on pseudo-labels, the models are prone to collapsing to the trivial solutions of predicting only one class to minimize the surrogate losses. Thus, to stabilize TTT, one heuristic is to balance the marginals of the model's predictions. Suppose for the moment that we have oracle access to the true label distribution in the target domain, $q(y)$. Then, a natural surrogate loss to minimize for balanced predictions is the Kullback–Leibler divergence between $q(y)$ and the average predictions of the model

$$\mathcal{L}^{(\mathrm{m})} = -\sum_{j}^{k} q(y_j) \log\left(\frac{1}{N}\sum_{i}^{N} p_\theta(y_j|x_i)\right), \tag{4}$$

which clearly discourages the model from converging to the trivial solutions. In practice, however, $q(y)$ and $\mathcal{L}^{(\mathrm{m})}$ are often unavailable and must be replaced by estimates $\hat{q}(y)$ and $\hat{\mathcal{L}}^{(\mathrm{m})}$. Li et al. (2023) uses the uniform prior $\hat{q}(y) = \frac{1}{K}$, but it is clear that this estimate is unsuitable for datasets with non-uniform prior, a common pitfall for TTT method as pointed out by Zhao et al. (2023). While standard benchmark datasets are often prepared to have uniform distribution by construction,

real data may deviate to long tail settings without warning under the TTT setting. A more practical and simple alternative is to use the initial pretrained model to estimate the label marginal, i.e. $\hat{q}(y) = \frac{1}{N} \sum_i^N p_{\theta_0}(y_j|x_i)$, where $\theta_0$ denotes the initial parameters of the pretrained model. Effectively, this constrains the deviations from the estimated initial label marginal.

**Thresholding.** Another intuitive idea to stabilize training is to filter out noisy supervision caused by incorrect pseudo-labels. One simple and widely used heuristic to obtain higher quality pseudo-labels is to use a threshold value to filter out potentially noisy labels: only the predictions with confidence higher than the threshold are used as pseduo-labels. Given a hyperparameter $\tau \in [0, 1)$ serving as threshold for the confidence, this approach replaces $\mathcal{L}^{(p)}(\theta)$ with the modified surrogate loss

$$\mathcal{L}^{(p)}(\theta; \tau) = -\frac{1}{N} \sum_i^N \sum_j^K \mathbb{I}\left[j = \max_k p_\theta(y_k|x_i)\right] \mathbb{I}\left[\max_k p_\theta(y_k|x_i) \geq \tau\right] \log p_\theta(y_j|x_i), \qquad (5)$$

where a common choice for $\tau$ is 0.9 (Sohn et al., 2020).

**Label Accumulation.** An alternative way to ensure high-quality pseudo-labels is to accumulate the historical pseudo-labels assigned to each sample and to regularize the pseudo-labels from abruptly deviating from the historical records. A simple way to achieve both goals is to smooth the pseudo-labels using exponential moving averages. Namely, we replace $\mathcal{L}^{(p)}(\theta)$ with the following loss

$$\mathcal{L}^{(p, s)}(\theta) = -\frac{1}{N} \sum_i^N \sum_j^K \hat{q}(y_j|x_i) \log p_\theta(y_j|x_i). \qquad (6)$$

where $\hat{q}(y_j|x_i)$ is estimated as the exponential moving average of all pseudo-labels assigned to $x_i$ over the training cycles. Note that accumulating historical pseudo-labels with an exponential moving average was first proposed by Laine & Aila (2017). They additionally utilize augmentation, bias correction, and penalize the deviation through square loss, whereas we only accumulate the pseudo-labels and use negative log-likelihood for surrogate loss instead.

**Mean Teacher.** Another common approach to regularize the pseudo-labels is through the mean-teacher scheme (Tarvainen & Valpola, 2017). This approach initializes two models: the teacher and the student with an initial model. The student adapts its parameters using the pseudo-labels provided by the teacher, and the teacher then updates its own parameters as the exponential moving average of the student's parameters.

## 5 STUDY ON COLLAPSE AND THE PRO METHOD

In this section, we assess the techniques introduced previously to adapt CLIP models on 14 standard vision datasets. Details on all the datasets considered in this work are provided in App A. We adapt only the linear classifier of ViT-B-32 to identify promising techniques. Empirically, we find that these techniques on their own either (1) do not prevent the collapses in test time training or (2) are insufficient to encourage significant improvement in models' accuracy. We then explore whether a combination of these techniques overcomes these shortcomings. From ablating over different design choices, we identify Pseudo-label Regularized Optimization (PRO), which uses the marginal consistency loss, cross-entropy of accumulated pseudo-labels, and test-time entropy losses jointly to achieve an average of 2% improvement in accuracy across all datasets when adapting the linear classifier of ViT-B-32 model.

Our starting point is the simple surrogate loss of Eq. 3. Throughout the work, we set the weight for test-time entropy $\lambda_e$ to a small value 0.1. We adapt the CLIP models for 200 epochs with a full batch size. Note that we are able to perform full gradient descent since we freeze the visual encoder, which enables us to feedforward all the images and store them as features. We choose to avoid stochastic gradient descent in the spirit of obtaining better estimates for the marginal consistency loss $\mathcal{L}^{(m)}$ and thereby stabilizing test-time training. We use the simple gradient descent optimizer with a 0.01 learning rate when adapting the linear layer and 0.1 momentum throughout this paper.

**Design protocol.** As discussed, many design choices could potentially help address the collapse. But it is not immediately clear how to identify good choices in a setting where access to test labels should not be assumed. Hence, we propose the following setup. First, to avoid meta-overfitting, we

leverage standard hyperparameters and practices from the literature and refrain from tuning them. Still, this does not help us identify better design choices. We make this decision by assessing the performance of each design choice on a large number of datasets (in total 14 different datasets) with all of its hyperparameters held fixed across datasets. From these experiments, we gather insights into how different regularization approaches actually perform. To test whether our insights generalize, we apply the best design choice identified to *four additional datasets*. Additionally, we experiment with updating a different set of parameters, that of the whole textual tower (Zhai et al., 2022). Finally, we also test whether the approach would generalize to a different visual backbone. Our protocol addresses the concerns of Zhao et al. (2023) as (1) hyperparameters are kept fixed to standard defaults, (2) we keep the type of the pre-trained model fixed but change the backbone or the parameter set we update, and (3) we consider a large number of datasets on which we base our design choices and multiple datasets for testing. In Appendix B, we also test PRO on a ResNet pre-trained with supervised learning.

**Exact MC Loss Prevents Collapse.** Experimentally, we find that minimizing $\mathcal{L}^{(m)}$ in addition to Eq. 3 using the ground-truth $q(y)$ is sufficient to stabilize TTT, resulting in a significant gain in accuracy (on average $3.1\%$). However, as the true label distribution $q(y)$ is often not available in realistic settings, we must consider minimizing an estimate $\hat{\mathcal{L}}^{(m)}$ instead.

**Estimated MC Loss Does Not Prevent Collapse.** While we find that using the uniform prior for $\hat{\mathcal{L}}^{(m)}$ also leads to significant gain ($2.7\%$ on average), we showed in App. C that, as one would expect, employing the uniform prior inevitably fails when the target label distribution is not uniform. Since the severity of class imbalances in the target domain is usually unavailable a priori, we use the soft label marginal $\hat{q}(y)$ instead. Empirically, we find that the soft label marginal largely prevents severe decline in accuracy, but on three datasets we still observe decrease in accuracy greater than $1\%$ and up to $2.8\%$.

*Discussion.* One can prevent collapses in test time training by enforcing a prior on the model's predictions. While we observe impressive gains using the true prior, knowledge of the label marginal of the target distribution may not be available in practice. On the other hand, the uniform prior is vulnerable to class imbalances at test time. Hence, we use the soft label marginal provided by the initial CLIP model as the label marginal estimate $\hat{q}(y)$ and adapt on the corresponding estimated loss $\hat{\mathcal{L}}^{(m)}$. However, we still observe few failure cases, which suggest that further regularization of test-time training is necessary.

**Thresholding And Estimated MC Prevent Collapse.** We experiment with adapting on the threshold loss with the common choice $\tau = 0.9$ (Sohn et al., 2020) without employing the MC loss. As expected, this still results in collapses due to the class imbalance in pseudo-labels with high confidence. We then use thresholding jointly with the MC loss and find that this prevents collapses across all datasets we considered and results in an increase in accuracy by $1.5\%$ on average.

**Label Accumulation And Estimated MC Prevent Collapse.** For the exponential moving average, we fix the decay rate to a conservative value $0.99$. We found that using this smoothed pseudo-label loss jointly with the marginal consistency loss also results in non-trivial gains ($1.6\%$ on average).

**Mean Teacher.** Throughout the experiments, we fix the decay rate of the exponential moving average to $0.999$ as suggested by Tarvainen & Valpola (2017). Applying mean teacher along with the MC loss results in only a $1.3\%$ increase in accuracy on average. We also find that applying mean teacher in addition to either technique above only worsens the results ($1.4\%$ gain with thresholding and $1.2\%$ gain with label accumulation). It seems that excessive regularization of the pseudo-labels diminishes useful information the model can extract from the new labels.

*Discussion.* We explored different approaches to regularize changes in pseudo-labels and found that they individually work with the marginal consistency loss in preventing collapses in TTT. However, when applied jointly, the additional regularizations seem to prevent potentially larger gains from test-time training. Moreover, we find that out of the three techniques, accumulating the pseudo-labels with a simple exponential moving average works the best empirically. Based on these results, we propose PRO-CLIP, which utilizes a combination of marginal consistency loss, label accumulation, and minimization of test entropy to stabilize test-time training. Figure 3a provides the performance of PRO on each dataset.

# 6 EXPERIMENTAL RESULTS

Thus far, we have experimented with a few combinations of techniques only by adapting the linear classifier of ViT-B-32 model. However, as pointed out by Zhao et al. (2023), one potential concern for test-time training method is that the design choices of PRO may meta-overfit to the particular settings it is developed for, such as the datasets, the tunable model parameters, and the CLIP model considered. To illustrate that PRO has not meta-overfitted to the aforementioned settings, we apply PRO to adapting on previously unseen datasets, tuning the textual encoder of CLIP, and tuning the larger ViT-L-14-336 model. When adapting the textual encoders, since we update considerably more parameters, we decrease the learning ten-fold to $0.001$ (which is a commonly used learning rate in deep learning, the default in Adam (Kingma & Ba, 2017)) while keeping all other hyperparameters the same. In addition, while we do not tune the hyperparameters of PRO, we still perform ablation studies on these hyperparameters near the employed values to demonstrate that PRO is resilient to changes in its hyperparameters. Finally, we compare PRO against prior work by Li et al. (2023) that also focuses on adapting CLIP models under the TTT setting. We focus our comparison with MUST since it was designed specifically to address the collapse of TTT for CLIP. Prior methods were shown to fail by Zhao et al. (2023), and our setting, adapting one single model to multiple data sets, can only exacerbate the issue.

## 6.1 ADAPTING TEXTUAL ENCODER WITH DIFFERENT VISUAL BACKBONE

**Adapting Textual Encoder** Here, we apply PRO-CLIP to update the entire textual encoder of ViT-B-32. This is suggested recently by Zhai et al. (2022), where the authors find that when tuning CLIP models, freezing the visual encoder and tuning only the textual encoder proves the most advantageous. Note that when tuning the textual encoders, we use the standard learning rate $0.001$ but do not change any other hyperparameters. The results are shown in Figure 3b. We find that the techniques we identified are successful in preventing collapses even when adapting the textual towers, suggesting that PRO-CLIP also did not overfit to the choice of which parameters are being updated. Additionally, we find that adapting the textual encoder generally results in higher accuracy ($2.2\%$ gain on average) when compared to adapting the linear classifier ($1.6\%$ gain on average), echoing the results obtained by Zhai et al. (2022) that finetuning textual encoder of CLIP is advantageous.

**Adapting ViT-L (Linear/Textual)** To provide additional evidence that PRO-CLIP works out of the box without changing the hyperparameters, we assess the performance of PRO-CLIP when adapting a larger CLIP model, e.g. ViT-L-14-336. We present the results in Figures 3c and 3d. The decline in model's accuracy when tuning the linear classifier is rare and the decreases are small in magnitude (up to $0.3\%$). When tuning the textual encoders, only one of the datasets experiences a drop in accuracy. In most other datasets, we also observe that tuning the textual encoder yields greater gain than tuning the linear classifier, suggesting that adapting the textual encoder of CLIP may be the preferred strategy.

## 6.2 NEW UNSEEN DATASETS

To demonstrate that PRO-CLIP does not overfit to the datasets we consider above, we apply it to additional datasets that were not considered when deciding the design choices for PRO-CLIP. We consider four additional dataset: ImagenetV2, UCF-101, DTD, and Entity13 (Recht et al., 2019; Soomro et al., 2012; Cimpoi et al., 2014; Santurkar et al., 2021), for which we find correspondingly $-0.1\%$, $+1.7\%$, $+3.3\%$, and $+7.8\%$ changes in accuracy from adapting the linear layer of ViT-B-32 using PRO-CLIP. The consistent gain suggests that the design choices and the hyperparameters of PRO-CLIP do not meta-overfit to the particular downstream prediction tasks we considered in the previous section.

## 6.3 ABLATIONS AND ROBUSTNESS TO HYPERPARAMETERS

In this section, we explore different options for the hyperparameters used in PRO. For the ablation study, we tune the linear layer of ViT-B-32 with the same setup as the main results by default: 200 epochs, $0.01$ learning rate, $0.1$ momentum, and $0.1$ for the weight $\lambda_e$ of the test entropy loss. We ablate over only one parameter at a time to study the sensitivity of PRO to such changes. Here,

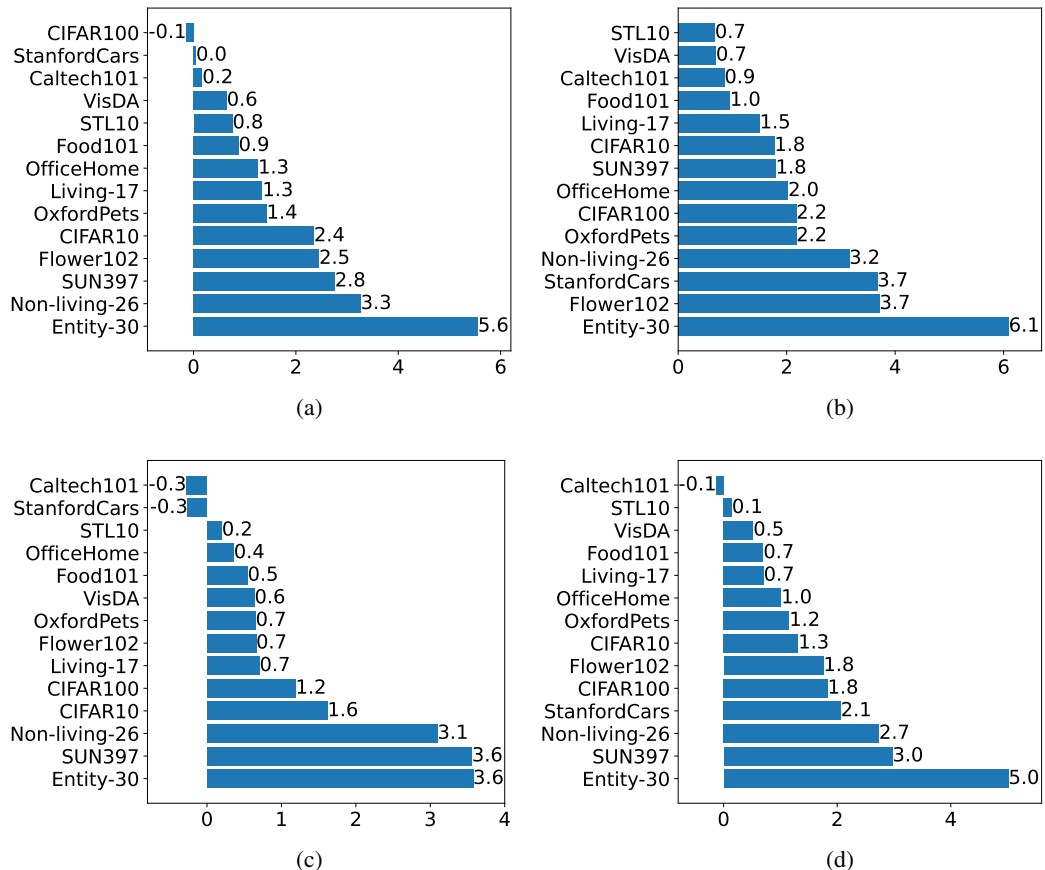

Figure 3: Gain in accuracy from applying PRO-CLIP. **(a, b)**: results from adapting the linear classifier (left) and textual encoder (right) of ViT-B-32. **(c, d)**: results from adapting the linear classifier (left) and textual encoder (right) of ViT-L-14-336.

Table 1: Comparison between MUST and our method using ViT-B-32. Note that in the original work MUST uses the uniform prior in Eq. 4. For a fair comparison we implemented a variant of MUST that uses the estimated prior from average predictions of the initial model. While the former performs well in DTD and UCF101, both of which have a uniform label marginal. On the other hand, our methods do not assume access to the exact label marginal and still improve on CLIP's zero-shot performances across all datasets.

|                  | ImageNetV2 | SUN397 | Food101 | DTD  | UCF101 |
|------------------|:----------:|:------:|:-------:|:----:|:------:|
| CLIP zeroshot    | 56.3       | 62.1   | 80.7    | 43.7 | 64.1   |
| MUST-uniform     | 55.1       | 50.9   | 73.1    | 49.9 | 68.1   |
| MUST-estimated   | 54.3       | 46.2   | 71.4    | 44.9 | 64.0   |
| PRO-CLIP (linear)| 56.2       | **64.8** | **81.6** | 47.0 | 65.7   |
| PRO-CLIP (textual)| **57.2**  | 63.9   | **81.6** | **48.7** | **67.1** |

we consider all 18 datasets, including the previously unseen datasets. For reference, the default hyperparameters achieve 2.0% increase in accuracy averaged across all 18 datasets.

**Ablation Over Number of Training Epoch.** Zhao et al. (2023) pointed out that the performance of TTT methods is sensitive to the number of training epochs. Indeed, without supervision, it is unclear how one chooses an optimal training epoch. In the TTT setting, the ideal training stabilizes after a certain number of epochs without a detrimental effect on the model's accuracy. Figure 4 shows that PRO indeed stabilizes as the number of epochs increases. In fact, the number of epochs we chose

(200) is far from the optimal epoch for datasets considered in this work. We also find that at the 1000th epoch, we do not observe collapse in any of the datasets.

**Ablation Over Learning Rate.** We try learning rates 0.02 and 0.005 and find that they correspondingly result in 1.8% and 1.6% increase in accuracy on average. Notably with a 0.02 learning rate, CIFAR-10 has a 1% drop in accuracy. This, combined with the ablation studies on the number of epochs suggests that lowering the learning rate and increasing the number of epochs may be a better hyperparameter choice.

**Ablation Over Momentum.** We experiment with momentum, 0.5 and 0.9, for which we find correspondingly 2.1% and 2.5% increase in accuracy on average. Similar to the results from ablating number of epochs, since we restrict ourselves from hyperparameter search in this work, the choice of hyperparameter is far from optimal, and the performance of PRO may be even greater.

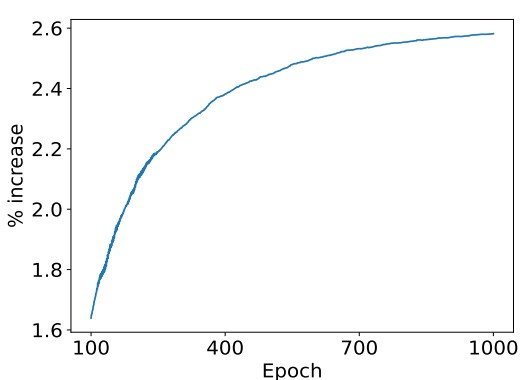

Figure 4: The average gain of PRO-CLIP as the number of training epochs increases. We notice that the accuracy smoothly increases on average. None of the 18 considered data sets collapsed.

## 6.4 COMPARISON WITH MUST

In Table. 1, we compare PRO-CLIP with Masked Unsupervised Self-Training (MUST), a recent work by Li et al. (2023) that also focuses on adapting CLIP models in the test-time training setting. Similar to PRO-CLIP, MUST also uses the marginal consistency loss, albeit with the uniform prior as the estimate for the target label distribution. In addition, they tune the visual tower and use the idea of consistency regularization to encourage the predictions of strongly augmented images to match with the pseudo-labels obtained from weakly augmented images. While Li et al. (2023) showed promising results in test-time training, it is not clear without access to labeled data how one selects the different hyperparameters that MUST employed for each dataset. Thus, for a fair comparison, we fix the hyperparameters of MUST across all datasets and apply MUST on ViT-B-32. We provided the employed hyperparameters of MUST in App. A. Table 1 demonstrates that without oracle access to the ideal hyperparameters for specific dataset, MUST may still result in worsening accuracy. In comparison, our method uniformly improves on all datasets with a fixed set of hyperparameters.

## 7 CONCLUSION

We investigated the common collapse failure in test-time training and considered various techniques to address it, borrowing ideas from prior works in the semi-supervised and test-time training settings. After a search for the best design choices, we propose PRO-CLIP, an unsupervised training scheme that utilizes marginal consistency, test entropy minimization, and label accumulation to prevent collapses and enable up to 7.8% improvement in accuracy of the ViT-B-32 CLIP model. We addressed the common pitfalls in test-time training methods as pointed out by Zhao et al. (2023) by applying PRO-CLIP on a wide range of new adaptation settings: new datasets, new set of model parameters to tune, and a larger ViT-L-14-336 CLIP model to adapt. The performance of PRO-CLIP under these new adaptation settings suggests that PRO-CLIP works out of the box without the need for hyperparameter tuning, which is difficult to perform under the test-time training setting. Throughout this work, we use the initial predictions of the CLIP model to estimate the marginal consistency loss. It is an interesting future direction to investigate whether the adapted CLIP model provides a better estimate for target label distribution, thereby achieving the gain we observe when using the exact label marginal. There are two main limitations of this work. First, while we considered a significantly large number of data sets, these may not be fully representative of many use cases that are relevant in practice, especially long-tail ones. Additionally, we focused on CLIP, but other foundation models could also have been interesting to test, especially language classifiers and visual-language generative models (Ramesh et al., 2022), where test-time training is less explored.

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

## A   More Details on Experiments

**Model details.**   In this work we considered ViT-B/32 and ViT-L-14-336 pretrained by OpenAI (Radford et al., 2021) using codebase provided by Cherti et al. (2022).

**Dataset details.**

- Caltech101 Li et al. (2022): A dataset of 12 categories. We use all 9k samples as unlabeled target data. There are only a few samples and the label distribution is highly non-uniform, so this datasets is particularly challenging for test-time training.
- CIFAR-10 Krizhevsky & Hinton (2009): A dataset of colored images of 10 items: airplane, automobile, bird, cat, deer, dog, frog, horse, ship, and truck. All 60k samples are used as unlabeled test data.
- CIFAR-100 Krizhevsky & Hinton (2009): A dataset of colored images of 100 items. All 60k samples are used as unlabeled test data.
- DTD Cimpoi et al. (2014): A dataset of 47 categories. We use all 6k samples from the three domains (train, val and test) as unlabeled target data.
- BREEDs Santurkar et al. (2021): The BREEDs benchmark contains four datasets:
    - Entity-13: A dataset of 13 categories. We use all 17k samples as unlabeled target data.
    - Entity-30: A dataset of 30 categories. We use all 16k samples as unlabeled target data.
    - Non-living-26: A dataset of 26 categories. We use all 70k samples as unlabeled target data.
    - Living-17: A dataset of 17 categories. We use all 46k samples as unlabeled target data.
- ImageNetV2 Recht et al. (2019): A new test data for the ImageNet benchmark with 1000 classes. All 10k samples are used as unlabeled test data.
- Oxford 102 Flower Nilsback & Zisserman (2008): A dataset of 102 categories. We use all 8k samples from the three domains (train, val and test) as unlabeled target data.
- Food101 Bossard et al. (2014): A dataset of 101 categories. We use all 101k samples as unlabeled target data.
- Oxford-IIIT Pet Parkhi et al. (2012): A dataset of 37 categories. We use all 7k samples as unlabeled target data.
- Office Home Venkateswara et al. (2017): There are four domains in Office Home dataset: art, clipart, product and real. We use all 4k samples from the product domain as unlabeled target data.
- Stanford Cars Krause et al. (2013): A dataset of 196 categories. We use all 8k samples as unlabeled target data.
- STL-10 Coates et al. (2011): A dataset of 10 categories. We use all 13k samples as unlabeled target data.
- SUN397 Xiao et al. (2010): A dataset of 397 categories. We use all 109k samples as unlabeled target data.
- VisDA Peng et al. (2017): A dataset of 12 categories. We use all 28k samples from the three domains (train, val and test) as unlabeled target data.
- UCF101 Soomro et al. (2012): A dataset of 101 categories. We use all 13k samples as unlabeled target data.

For prompts and class names, we generally follow the choice made by Radford et al. (2021). For ImageNetV2, we use the 7 best prompts identified by Radford et al. (2021). For other datasets without templates available in prior works, we use the 80 templates for ImageNet as default templates.

**Training details.** When tuning the textual encoder of CLIP, since the memory requirement of tuning with all prompts at once is too overwhelming, at each epoch, we randomly select one of the prompts for each class for training. At evaluation, we again take the average over these prompts for prediction.

**Hyperparameters** To fairly evaluate MUST (Li et al., 2023), we fix the hyperparameter across all datasets. The hyperparameters are: 32 for mask patch size, 0.9998 for the decay rate of the mean teacher, 0.1 for mask ratio, 0.5 for the weight of the global-local feature alignment loss, 1.0 for the marginal consistency loss, 0.7 for pseudo-labeling threshold, $10^{-5}$ for learning rate, and 40 for the number of the training epochs.

## B    ADAPTING TO SHIFTS ON RESNET

The test time training strategy proposed in this work is also applicable to models more traditionally trained to predict a predetermined set of classes. We consider the distribution shift between CIFAR-10 (Krizhevsky & Hinton, 2009) and CIFAR-10-C (Hendrycks & Dietterich, 2019b). We take the ResNet-34 backbone (He et al., 2016) pretrained on ImageNet and train a linear layer on top for CIFAR-10. We train the model for 20 epochs with 200 samples per batch using a simple stochastic gradient descent optimizer with 0.1 learning rate, 0.9 momentum and $10^{-4}$ weight decay.

We apply PRO to adapt the trained model on CIFAR-10-C data with the corruption method fog and severity level 1. We use the same hyperparameter choice as in the main text and use 0.01 for learning rate. We find that PRO provides minor improvement in accuracy (0.1% gain).

## C    FAILURE CASE OF UNIFORM PRIOR

To demonstrate a failure mode of using uniform prior for marginal consistency loss, we generate a new dataset by sampling data from CIFAR-10 to have large class imbalances. We generate the new label distribution randomly from the Dirichlet distrbution with parameter $\alpha = 1$: the new label marginal is sampled as $q_t(y) \sim \text{Dir}(\beta_y)$ with $\beta_y = \alpha p(y)$, where $p(y)$ is the original label marginal, which is uniform for CIFAR-10. Note that smaller $\alpha$ makes the shift in label marginal more severe.

We adapt the linear classifier of ViT-B-32 on the simple combination of pseudo-labeling loss and the marginal consistency loss $\mathcal{L}(\theta) = \mathcal{L}^{(\text{p})}(\theta) + \hat{\mathcal{L}}^{(\text{m})}$. We again use the simple gradient descent optimizer with learning rate 0.01 and momentum 0.1, and we train for 50 epochs. Under this setting, we compare the performance of using the label marginal estimated by model's prediction and that of using the uniform prior to estimate $\hat{\mathcal{L}}^{(\text{m})}$. The former manages to prevent collapse with 90% accuracy, whereas the latter with uniform prior drops to 78% accuracy.

## D    ADDITIONAL ABLATION STUDIES

We continue to explore different options for the hyperparameters used in PRO. Again we tune the linear layer of ViT-B-32 with the same setup as the main results by default: 0.01 learning rate, 0.1 momentum, and 0.1 for the weight $\lambda_e$ of the test entropy loss. Note the default hyperparameters acheive 2.0% increase in accuracy on average.

**Ablation Over Weights for Label Sharpening.** We try setting the weight of test entropy loss $\lambda_e$ to be 0.0 and 0.5 and we find correspondingly 1.8% and 1.6% increase in accuracy on average. The worsening results with $\lambda_e = 0.0$ shows the test entropy loss indeed helps adapting CLIP. However, with higher weights $\lambda_e = 0.5$ we find that some datasets like CIFAR-100 see a drop in accuracy by 5%, suggesting that a high value of $\lambda_e$ may derail test-time training.

**Ablation Over Weights for Marginal Consistency.** Note that we can introduce a weight $\lambda_p$ to the MC loss of Eq. 4 similar to $\lambda_e$ (by default $\lambda_p = 1$). We try 0.5 and 1.5 for $\lambda_p$ and find correspondingly 1.4% and 2.0% increase in accuracy on average. Notably CIFAR-100 and Stanford Cars saw 2% and 1% drop in accuracy correspondingly. Note that both values prevent catestrophic collapses, but high enough weight for the MC loss seems to be helpful for test-time training overall.

