# OpenReview forum: "PRO: Pseudo-label Regularized Optimization on Unlabeled Test Data"
_ICLR.cc/2024/Conference — Submitted to ICLR 2024_

### Official Review · Reviewer_JWjE · 2023-10-30

**Soundness:** 2 fair
**Presentation:** 2 fair
**Contribution:** 2 fair
**Rating:** 3
**Confidence:** 3

**Summary:**

This paper studies test time training methods, which may catastrophically collapse sometimes. This paper empirically tests various existing test time training methods that try to prevent collapse and re-evaluate their effectiveness and validity. By combining those methods based on the extensive insights for several datasets, experiments show that the proposed method can safely improve performance with test time training in most of datasets.

**Strengths:**

- Related works are re-investigated thoroughly and lessons from them are provided.

**Weaknesses:**

- Just combining existing methods, by trying everything and finding a good combination, might not be considered novel.
- While this paper empirically validates which setups/parameters are useful for test time training, it does not provide theoretical understanding on why they do not collapse.
- Details of algorithm (PRO) are missing. It is unclear how the loss functions are combined.

**Questions:**

- Can all experiment/analysis results of section 5 be included in appendix? While Section 5 contains various insights from experiments, full experiment setup and results are omitted.
- What’s the failure mode of the suggested method other than class imbalance?
- Can it be shown that PRO is effective in setups other than CLIP?

---

> ### Author Response · Authors · 2023-11-21
>
> We thank the reviewer for their suggestions to improve the paper. We provide some additional clarifications to the questions below:
>
> >**While this paper empirically validates which setups/parameters are useful for test time training, it does not provide theoretical understanding on why they do not collapse**
>
> We note that imposing a penalty against deviance from the initial label distribution intuitively discourages collapsed solutions. The primary aim of this paper is to conduct an empirical investigation to substantiate this intuition and explore methods useful for test-time training. The potential for theoretical exploration into this aspect is left for future works.
>
> >**Details of algorithm (PRO) are missing. It is unclear how the loss functions are combined**
>
> We will add equations detailing each loss function in the updated draft.
>
> >**Can all experiment/analysis results of section 5 be included in appendix? While Section 5 contains various insights from experiments, full experiment setup and results are omitted**
>
> We will expand on the discussion on the full experiments in the appendix in the updated version.
>
> >**What’s the failure mode of the suggested method other than class imbalance?**
>
> As shown in Figure. 3, we do not observe any significant failure mode of the suggested method.
>
> >**Can it be shown that PRO is effective in setups other than CLIP?**
>
> Our primary focus lies in understanding the distinctive distribution shifts associated with the adaptation of CLIP models in zero-shot tasks. While PRO may demonstrate effectiveness in handling other shifts and architectures, we consider these settings more appropriate for exploration in future works.

---

### Official Review · Reviewer_Bw6d · 2023-10-31

**Soundness:** 2 fair
**Presentation:** 3 good
**Contribution:** 2 fair
**Rating:** 3
**Confidence:** 4

**Summary:**

The authors note that prior approaches to test-time adaptation, which involve training on unlabeled test data, are often challenging to execute due to the sensitivity to hyperparameter selection, with suboptimal choices potentially leading to reductions in accuracy. To alleviate this issue, this paper introduces Pseudo-label Regularized Optimization (PRO), a strategy that combines various established techniques to mitigate the mentioned difficulties.

**Strengths:**

- This paper is well-written and easy to understand. It highlights the existing challenge where a neural model could fail due to the selection of suboptimal hyperparameters in test time training settings.
- The proposed PRO method is simple and demonstrates its efficacy in tuning CLIP models for visual data tasks.

**Weaknesses:**

- My primary concern is with the novelty of the proposed method PRO. The issue that the paper addresses was initially noted in [1], and it appears that PRO is essentially a combination of several pre-existing approaches.
- The authors have neglected to benchmark PRO against other related test-time adaptation methods employing semi-supervised learning strategies, such as self-training [2].
- The efficacy of PRO seems limited to specific architectures; it yields noticeable improvements when applied to VIT models but is virtually ineffective with ResNet architectures, resulting in a marginal improvement of only 0.1% (See Appendix B).

[1] Hao Zhao, Yuejiang Liu, Alexandre Alahi, and Tao Lin. On pitfalls of test-time adaptation. In ICLR 2023 Workshop on Pitfalls of limited data and computation for Trustworthy ML, 2023.

[2] S. Sinha, P. Gehler, F. Locatello, and B. Schiele. Test: Test-time self-training under distribution shift. In 2023 IEEE/CVF Winter Conference on Applications of Computer Vision (WACV).

**Questions:**

See weaknesses.

---

> ### Author Response · Authors · 2023-11-21
>
> We would like to thank the reviewer for their comments. We are glad that you appreciate the simplicity of our method.
>
>
> >**The authors have neglected to benchmark PRO against other related test-time adaptation methods employing semi-supervised learning strategies, such as self-training [2]**
>
> We thank the reviewer for bringing relevant works to our attention. However, we note that many test-time adaptation methods, including the one reviewer suggested, are designed for non-CLIP models. At a fundamental level, the distribution shifts addressed in these works differ substantially from those encountered in our context. ResNet models, for example, are trained on supervised classification tasks with training data resembling the test data. CLIP models, however, were trained via contrastive losses on data that may be significantly different or irrelevant to the downstream data we aim to adapt to. Additionally, it is unclear whether the adaptation should be applied to either or both of the visual and textual towers of CLIP, and improbable that the hyperparameter choices are transferable.
>
> We remark that comparison of PRO against a test-time adaptation method developed for CLIP (MUST) is provided in Table 1, where we find significant advantage in PRO over MUST. We additionally implement test-time prompt tuning (TPT) of Shu et. al [1] and found that on ViT-B/32, TPT results in 43% and 64% in accuracies for DTD and UCF-101 datasets, which are correspondingly 6% and 3% lower than the performance of PRO-CLIP with textual layer (see Table. 1)
>
> >**The efficacy of PRO seems limited to specific architectures; it yields noticeable improvements when applied to VIT models but is virtually ineffective with ResNet architectures, resulting in a marginal improvement of only 0.1% (See Appendix B).**
>
> We appreciate the reviewer's keen observation. It's important to highlight that the distribution shifts examined in Appendix B with the ResNet architecture differ significantly from those addressed in the main text with CLIP. While we acknowledge that PRO might be more suited for the latter scenario, we emphasize that as foundation models like CLIP gain widespread use, the latter setting becomes increasingly relevant.
>
> [1] Shu et. al. Test-Time Prompt Tuning for Zero-Shot Generalization in Vision-Language Models. In Advances in Neural Information Processing Systems (2022).

---

### Official Review · Reviewer_Bx9u · 2023-11-01

**Soundness:** 2 fair
**Presentation:** 2 fair
**Contribution:** 1 poor
**Rating:** 3
**Confidence:** 4

**Summary:**

Considering that a pre-trained model performs poorly in an unknow test environment, practitioners hope to use these unlabeled test data to improve model performance in this specific domain, also called **test-time training (TTT)** task.
The core of this TTT task lies in how to performhigh-quality pseudo labelling, that is, while improving the performance of the target domain, we must get rid of these following constraints: *1)-the hyperparameters of the method*, *2)-the type of pretrained models*, and *3)-the nature of the distribution shifts*.
For this purpose, this paper proposed **pseudo-label regularized optimization (PRO)**, a strategy that comprise of a set of regularization method for TTT to avoid catastrophic collapse and improve model performance.
Experiments on VIT-B-32 and ViT-L-14-336 models confirmed the success of the method.

**Strengths:**

[1] The **starting point** of this research is **great**.
Especially now that the pre-training paradigm is popular, how to use unlabeled data to improve the zero-shot capability of the model in the target domain is deemly important to the AI ​​community.

[2] The entire paper is **well laid out** and **easy to understand**.
The formulas and diagram in the *Sec 3. Problem Setup* allowed me to quickly understand the problems that the TTT task wanted to solve.

[3] The **experiment** appears to be **well designed** and **executed**.

**Weaknesses:**

1. Pro is a combination of existing mature methods, with **incremental originality**.
The methods described in Sec.4 are all existing methods and seem to be background descriptions.
I did not find a detailed description of PRO. As the authors write in the methods and conclusions, PRO is a cherry-picked combination of existing heuristic methods through ablation experiments.

2. The **experimental results** of the paper are **not convincing enough**.
First of all, PRO-CLIP in Figure 3 has little performance gains on commonly used image datasets.
Secondly, PRO is a cherry-picked combination of existing methods. Without the support of convincing experimental results, it is difficult to verify its effectiveness.
The authors could consider performing extensive experiments in additional text modalities, which I believe is fully achievable with a CLIP-based approach.

3. This article seems to be a relatively complete **experimental report**.
The author can choose to submit it to the official-themed workshop.

4. Some trivial tips:

    - You can consider displaying the experimental results in the form of figures and tables to facilitate readers to understand more quickly what the experiment is trying to express/prove.

    - The sketch of PRO in Fig.2 is not intuitive enough. Let people understand the composition of the entire method as soon as possible.

    - ‘’Text Tower'' needs a clear explanation.

**Questions:**

Please refer to the above-mentioned **Weakness** Part.

---

> ### Author Response · Authors · 2023-11-21
>
> We would like to thank the reviewer for their constructive feedback on our work. We are glad that you appreciate the clarity of our presentation and find merit in our experimental design.
>
> >**The experimental results of the paper are not convincing enough. First of all, PRO-CLIP in Figure 3 has little performance gains on commonly used image datasets**
>
> We'd like to highlight that PRO-CLIP leads to an average accuracy improvement of 2.5%, with a maximum enhancement of 6.1% for CLIP. Unlike prior works, we do not perform hyperparameter selection by peeking at the resulting accuracies of individual datasets. Instead, we adopt a consistent set of hyperparameters and conduct careful experiments to avoid meta-overfitting, thereby ensuring the robustness of our approach. We substantiated the robustness of our method in Sec. 6.2, where we find that PRO-CLIP with the same hyperparameter selection results in significant gain (up to 7.8% increase in accuracy) on additional, previously unseen datasets.
>
> >**Some trivial tips**
>
> We thank the reviewers for these suggestions and will incorporate them in the updated draft.

---

> ### Comment · Reviewer_Bx9u · 2023-11-22
> **Respont to the Authors**
>
> I would like to extend my appreciation for the authors' response and acknowledge their efforts in this work.
> However, it is regretely that the method innovation and performance improvement claimed by the authors do not convince me.
>
> 1. In terms of **method innovation**:
>
>     I still believe that this method share a high degree of similarity within other methods in the same sub-field.
>
>     The relationship between them more like different combinations of functional/module levels to handle different test scenarios.
>
> 2. In terms of **performance improvement**:
>
>     Based on a model as large as CLIP, the proposed method solely achieves a tiny gains (<= 3.5) in average absolute accuracy on almost all adopted image classification datasets.
>
>     A considerable part of the datasets are common and simple (such as CIFAR100, Flower102, Caltech101, CIFAR10, etc.).
>
>     Most importantly, these datasets are already present in the pre-training phase of CLIP, so even changes in certain hyperparameters may alter their performance gains. In-depth ablation analysis in this regard is also worth discussing, which will help demonstrate the effectiveness of this approach.
>
> Finally, the absence of experiments on textual datasets still leaves me concerned that the applicability of this approach is limited.

---

### Official Review · Reviewer_A2sn · 2023-11-06

**Soundness:** 3 good
**Presentation:** 3 good
**Contribution:** 3 good
**Rating:** 6
**Confidence:** 3

**Summary:**

The paper introduces PRO (Pseudo-label Regularized Optimization) as a solution to the common problem of model accuracy declines during test-time adaptation using pseudo-labeling techniques. PRO combines ideas from test-time training and semi-supervised learning, offering an effective approach to improve model adaptation without the need for hyperparameter tuning. Specifically, PRO employs a combination of a pseudo-label based surrogate loss and a standard surrogate loss, with a regularization parameter, to guide the model's adaptation. Experimental results on 18 datasets demonstrate that PRO enhances the accuracy of the ViT-B-32 model by an average of 2.5% and up to 6.1%, showcasing its practical utility in addressing this challenge.

**Strengths:**

Originality:

The proposed method, PRO, attempts to address the typical decline in model accuracy during test-time adaptation by using heavily regularized pseudo-labeling methods which I see as a novel solution.

Quality:

The paper conducts rigorous experiments with 18 datasets to illustrate the challenges of model adaptation and demonstrate the effectiveness of the PRO method in preventing model accuracy collapses by utilizing different existing methods related to effective pseudo labeling that are marginal consistency, test entropy minimization, and label accumulation.

Clarity:

The paper is clear in presenting its problem statement, proposed method, and experimental findings, with nice plots that highlight the research's key insights and contributions.

Significance:

This research holds significant practical value for real-life applications where acquiring labels is expensive but acquiring unlabeled examples is cheap. This work addresses a critical issue in the adaptation of large pre-trained models towards specific domains given few labeled examples.

**Weaknesses:**

- My biggest concern is the lack of mean and standard deviation in the reported results to see if the results are significant. This would require repeating the experiments multiple times.

- My other concern is the novelty seems limited as it simple utilizes multiple existing methods for regularizing pseudo-labeling marginal which are consistency, test entropy minimization, and label accumulation.

- Another concern is the lack of generalization to other tasks like image segmentation and text-related problems which have become high in demand these days.

- An additional concern is that the authors are using ImageNet pretrained model which has been pre-trained on millions of images that share similar classes as the ones given by the unlabeled set - to me it doesn't seem like this work would strictly fit within the low-data learning framework.

**Questions:**

Please address the weaknesses above.

---

> ### Author Response · Authors · 2023-11-21
>
> We are thankful for Reviewer A2sn’s thoughtful comments. We are glad that you appreciate the experiments and the writing.
>
> >**My biggest concern is the lack of mean and standard deviation in the reported results to see if the results are significant. This would require repeating the experiments multiple times**
>
> We wish to emphasize that our approach lacks inherent randomness, as a result we are not able to calculate standard deviation. We remark that the experimental results show that our proposed method consistently enhances the accuracy of CLIP over 18 datasets, and hope that this addresses the reviewer’s concern on the robustness of our method.
>
> >**Another concern is the lack of generalization to other tasks like image segmentation and text-related problems which have become high in demand these days**
>
> We agree with the reviewer that these are essential tasks for test-time adaptation. As an initial evaluation of the proposed method, we have chosen to concentrate on image classification tasks as a starting point. The investigation of appropriate methods for test-time adaptation in other tasks is deferred to future work.
>
> >**An additional concern is that the authors are using ImageNet pretrained model which has been pre-trained on millions of images that share similar classes as the ones given by the unlabeled set - to me it doesn't seem like this work would strictly fit within the low-data learning framework**
>
> We want to clarify a confusion around this. We do not claim that we are in the low-data learning framework. Instead, our primary focus is on adapting a pretrained model using unlabeled downstream data.

---

> > ### Comment · Reviewer_A2sn · 2023-12-05
> > **Response to Authors**
> >
> > I appreciate the response to my concerns. However, as others and I mentioned, there is a novelty concern here as the method greatly overlaps with existing ones.
> >
> > I will keep the score as is.

---

### Author Response · Authors · 2023-11-21

We appreciate the reviewers' careful evaluation of our work and thoughtful feedback. We are glad that reviewers appreciate our experiments (A2sn, Bx9u, Bw6d, JWjE) and find the papers well written (A2sn, Bx9u, Bw6d). While the initial reviews are mixed, we believe the concerns raised can be effectively addressed in this round of rebuttal.

A common concern revolves around the novelty of the proposed method. We would like to highlight that while our methods share similarities with existing approaches, there are differences that are key to the success of our method.
Marginals Regularization: we opt for regularization based on the estimated label marginal, in contrast to previous methods that use a uniform prior. This distinction is critical as regularizing towards uniform priors can lead to a significant (more than 12%) drop in accuracy, as demonstrated in Appendix C, when the true label marginals sufficiently deviate from the uniform prior..
Label Accumulation: In comparison to Laine & Aila 2017 [1], where a subroutine resembling label accumulation exists, we use accumulated pseudo-labels differently. Specifically, they apply L2 loss to pseudo-labels, perform bias correction, and introduce a ramp-up weight for unsupervised loss. Our preliminary experiments indicate that their setup, designed for a semi-supervised setting, is unsuitable for test-time training. We simplify the procedure by training via cross-entropy instead without other modifications.
We apologize for the lack of clarifications regarding the novelty of our method, and we hope that this addresses your concern. Please let us know if there are any remaining concerns. We will be happy to answer any further questions you may have.

[1] Samuli Laine and Timo Aila. Temporal ensembling for semi-supervised learning. In International Conference on Learning Representations (2017).

---

### Meta-Review · Area_Chair_bcAx · 2023-12-08

**Metareview:**

This paper focuses on the problem of collapses in test-time adaptation. A method is proposed that employs a set of regularizations during test-time training to prevent collapses.

During the rebuttal, the authors did not provide rebuttals to address the concerns raised by three reviewers, including originality, effectiveness, and insufficient evaluation. Therefore, the AC recommends rejecting this paper.

**Justification For Why Not Higher Score:**

The authors did not provide rebuttals to address the concerns raised by three reviewers, including originality, effectiveness, and insufficient evaluation.

**Justification For Why Not Lower Score:**

NA

---

### Decision · Program_Chairs · 2024-01-16

Reject